# Construction of the Short-Form Thai-Home Fall Hazard Assessment Tool (Thai-HFHAT-SF) and Testing Its Validity and Reliability in the Elderly

**DOI:** 10.3390/ijerph19095187

**Published:** 2022-04-24

**Authors:** Charupa Lektip, Sarawut Lapmanee, Rewwadee Petsirasan, Kanda Chaipinyo, Saifon Lektip, Jiraphat Nawarat

**Affiliations:** 1Department of Physical Therapy, School of Allied Health Sciences, Walailak University, Nakhon Si Thammarat 80160, Thailand; charupa.le@wu.ac.th; 2Movement Sciences and Exercise Research Center, Walailak University, Nakhon Si Thammarat 80160, Thailand; 3Department of Basic Medical Sciences, Faculty of Medicine, Siam University, Bangkok 10160, Thailand; sarawut.lap@siam.edu; 4Faculty of Nursing, Prince of Songkla University, Songkla 90110, Thailand; rewwadee.p@psu.ac.th; 5Faculty of Physical Therapy, Srinakharinwirot University, Nakhonnayok 26120, Thailand; kanda@swu.ac.th; 6Seka Hospital, Bueng Kan 38150, Thailand; saifonphonpomrach@gmail.com

**Keywords:** validity and reliability, home environments, falls, elderly

## Abstract

The Thai-Home Fall Hazard Assessment Tool (Thai-HFHAT) was developed to identify the fall risk among the elderly arising from their home environment. However, it is more time consuming for large items. Therefore, this study developed a short-form of Thai-HFHAT (Thai-HFHAT-SF). In phase I, we developed the Thai-HFHAT-SF by performing a confirmatory factor analysis (CFA) of 450 rural elderly people. In phase II, a total of 105 participants; 50 elderly people, 50 caregivers, and 5 village health volunteers (VHV) were recruited to examine the reliability of the Thai-HFHAT-SF. Intra-class correlation coefficient (ICC) was used to analyze the inter-rater and test–retest reliability. Factor analysis selected 28 out of the 69 original Thai-HFHAT items in 4 components: indoor area, garage, outdoor areas, and risky spots/areas including pets. The factor loading was 0.67, 0.60, 0.32, and 0.31 in each component. The fitness index indicated that this model was fit (*χ*^2^/df = 1.38, goodness-of-fit Index (GFI) = 0.988, adjusted goodness-of-fit index (AGFI) = 0.970, standardized root mean square residual (SRMR) = 0.030, and root mean square error of approximation (RMSEA) = 0.029). The inter-rater reliability of the Thai-HFHAT-SF was 0.82 (95% CI: 0.71–0.89). The test–retest reliability was 0.77 (95% CI: 0.60–0.87) for the older person group, 0.85 (95% CI: 0.73–0.91) for the caregiver group, and 0.60 (95% CI: 0.29–0.77) for the VHV group. The new 28-item scale focused on home fall hazards and can be conducted in 10–15 min. Thai-HFHAT-SF is suitable for home hazards assessment among elderly in Thailand.

## 1. Introduction

Falls are a major public health problem in the elderly. The WHO Global Report on Falls Prevention in Older Age estimates that approximately 28–35% of adults aged over 65 years will report at least one fall each year [1]. Thailand’s Department of Disease Control has predicted that during 2017 to 2021, falls among the Thai elderly will account for 27% of deaths in the elderly, resulting in an estimated death rate due to falls among Thai elderly of 50 per 100,000 population [2]. The physical impact of falls in the elderly consists of mild to severe injuries with additional impacts on mental health. The elderly who suffered from serious injuries will need to stay in the hospital for a long time, and the impact on the economy and society caused by medical expenses while in the hospital will increase. For this reason, if a fall can be prevented, it will help improve the quality of life of the elderly and help reduce costs from falls [3,4]. The factors contributing to falls were classified as biological, behavioral, social, economic, and environmental factors [1].

The environmental factors include home and environmental hazards, such as insufficient lighting, uneven flooring, and loose mats. Steep or narrow stairs/steps were also important factors that caused falls in the elderly [5]. Falls can be reduced by using an appropriate home hazard screening instrument and making changes to the surroundings [6]. A wide variety of fall risk screening tools are currently available; each has different methods or types of assessment, scores, duration of screening performed by patients or health professionals, and languages. Among these tools, the Home Falls and Accidents Screening Tool (HOME FAST), a 25-question screening tool, takes psychometric properties into account and is widely accepted for screening the risk of falls in elderly [7]. In 2011, a self-reported home falls hazard screening tool (HOME FAST-SR) was developed from the 25-question HOME FAST by Hassani et al. This tool consists of 87 questions that are filled out by the elderly to screen for fall risk, and it showed moderate agreement with the HOME FAST. Correlation between scores from the HOME FAST-SR and a fall accident within the 6 months before the HOME FAST-SR was administrated and determined (*p* = 0.0001) [8]. However, both assessments have limitations in geographic, cultural, and architectural design differences, making some assessments unavailable among the elderly in Thailand.

In Thailand, the Thai Home Falls Hazards Assessment Tool (Thai-HFHAT) has been developed and accepted as the only reliable home hazard fall risk screening instrument. This tool has 69 questions and takes about 45 min to administer. It has been designed in such a way that the elderly can fill it out with or without help from Thai village health volunteers (VHV). The ICC of inter-rater reliability and test–retest reliability for the Thai-HFHAT was 0.87 (95% CI: 0.78–0.93) and 0.78 (95% CI: 0.58–0.89), respectively [9]. The cutoff point of the Thai-HFHAT was 18, which predicted the risk of falls due to home hazards in the elderly. However, the Thai-HFHAT is complex and time consuming, and it is difficult for the elderly to precisely complete all the questions. This is an important shortcoming of the Thai-HFHAT, limiting its routine use. Therefore, the purpose of this study is aimed at evaluating the construct validity of the Thai-HFHAT to further develop a model for a short-form version with a reduced number of items for the Thai elderly. The reliabilities of the instrument were also assessed.

## 2. Materials and Methods

This cross-sectional study was conducted at Nakhon Si Thammarat, Thailand from 26 June 2021, to 15 August 2021, and consisted of 2 phases. A confirmatory factor analysis (CFA) of the 69-question Thai-HFHAT was performed in phase I. A short-form of the Thai-HFHAT (Thai-HFHAT-SF) based on the original version was developed, and reliabilities of the Thai-HFHAT-SF were evaluated in phase II.

### 2.1. Phase I

The survey method examined the achievement of fall risk screening data of 450 elderly [9] who were recruited to participate and fulfill the sample size conducted in phase I. Because the suitable samples were from 200 or more, the study used a confirmatory factor analysis (CFA) [10]. We used the 69-question Thai HFHAT for data collection. The Thai-HFHAT was constructed by the researchers to assess the fall risk among the elderly in Thailand. It was composed of relevant variables collated from Thai and international literature. These variables were reviewed and discussed in a focus group with the experts. The researchers then constructed a self-reported home hazards assessment instrument based on the review. The inter-rater reliability and test–retest reliability of the 69-question Thai-HFHAT were 0.87 (95% CI: 0.78–0.93) and 0.78 (95% CI: 0.58–0.89), respectively. The elderly were asked to be screened for fall risk according to 9 areas in their house: 6 items for a living room, 7 items for a kitchen, 13 items for a bathroom, 10 items for a bedroom, 10 items for stairs, 7 items for a garage, 14 items for an outdoor area, 1 item for shoes, and 1 item for a pet, including the incidence of fall, was followed up and collected in 1 year. After the data collection from the 69-question Thai-HFHAT, we developed the Thai-HFHAT-SF according to the predictive model of fall risk from CFA. The result of CFA reduced the fall risk items from 69 to 28 and covered 4 main factors: 12 items for an indoor area, 4 items for a garage, 7 items for an outdoor area, 4 items for risky spots/areas, and 1 item for pets. Then, the content validity of the Thai-HFHAT-SF was assessed by 3 experts with over 3 years of experience; an architectural expert, and 2 physical therapy experts. A content validity index (CVI) included the item content validity index (I-CVI) and the content validity for scale (S-CVI) given by the scholars, and was used to describe the content validity of the instrument.

### 2.2. Phase II

The cross-sectional survey was designed to investigate the content validity and the inter-rater reliability of the Thai-HFHAT-SF. The test–retest reliability was evaluated using a prospective design that all participants answered (Thai-HFHAT-SF) and 2 weeks later, the Thai-HFHAT-SF questions were repeated. This study was approved by the Institute Review Board Walailak University (IRB reference no. WUEC-20-302-01) before commencement. This phase was conducted in the elderly, caregivers, and village health volunteers (VHV) recruited from Tha Sala District, Naknon Si Thammarat by convenience sampling method. The inclusion criteria of the elderly were both genders and aged ≥ 60 years with fluency in Thai who were willing to participate in this study. In addition, caregivers aged < 60 years (20–50 years) who spent most of their time taking good care of the elderly and VHV were also included in this study. The elderly people who were unable to perform daily activities and those who exhibited the symptoms of early-onset dementia were excluded. A number of 50 elderly were included and eligible for the evaluation of inter-rater and test–retest reliabilities [11]. In this study, these 2 tests were used to examine the level of reliability for Thai-HFHAT-SF between the 3 groups being 50 study subjects, 50 caregivers (healthy/young persons), and 5 village health volunteers (VHV). These 3 groups of participants were chosen to determine whether each group of subjects can be substituted for the other when assessing the fall hazards in the event the elderly are unable to complete the instruments by themselves.

The procedure of test–retest reliability of the Thai-HFHAT-SF was performed twice by the elderly during the first home visit and 2 weeks after the first visit, as recommended by Marx et al. [12]. Moreover, inter-rater reliability was also performed between the elderly, caregivers, and VHV. Intra-class correlation coefficient (ICC) was used to evaluate the test–retest reliability of the instrument. In addition, potential bias during these assessments was prevented by instructing each group of subjects to independently fill out the instrument. The score of Thai-HFHAT-SF of each group was evaluated with inter-rater reliability.

### 2.3. Statistical Analysis

Data were analyzed using the statistical package for the social sciences (SPSS), version 22.0 for Windows (IBM Corporation, New York, NY, USA). Confirmatory factor analysis for categorical data (CFA, LISREL 8.72 software, Scientific Software International, Inc., Lincolnwood, IL, USA) was also performed.

#### 2.3.1. Phase I

Demographic data were analyzed using descriptive statistics. The assumptions of factor analysis were to meet the criteria of: (i) univariate outlier standardized scores must be greater than or equal to 3.30 (|z| ≥ 3.30) and multivariate outlier determined from the Mahalanobis distance value must be (*p* < 0.001), (ii) normality determined from skewness (≤2) and kurtosis (≤7), (iii) suitability of data determined from Kaiser–Meyer–Olkin: KMO must be close to 1, and (iv) correlation determined from chi-square must be (*p* < 0.05). The secondary order CFA was performed based on the correlation matrix to evaluate the construct validity of the model with 9 components. Maximum likelihood estimation (ML) was conducted to determine values for the parameters in the examination of construct validity. Structural equation modeling (SEM) was used to determine the fitness of the model using the following statistics: Chi-square (*χ*^2^), chi-square to df ratio (*χ*^2^/df), GFI, AGFI, CFI, SRMR, and RMSEA. A comparison of component weights and empirical data was carried out to determine the weight of four factor components.

#### 2.3.2. Phase II

The content validity index (CVI) was classified as follows: 1.00 was classified as not relevant, 2.00 as somewhat relevant, 3.00 as quite relevant, and 4.00 as relevant [13]. The item content validity (I-CVI) was determined by the number of experts giving a score of 3 or 4 to each question, and divided by the total number of experts. I-CVI > 0.67 was classified as having an acceptable agreement, >0.80 as a good agreement, and 1.00 as an excellent agreement [14]. The content validity for scale (S-CVI) was the ratio of the number of questions with a score of 3 or 4, divided by the total number of questions. An S-CVI > 0.80 was classified as having good content validity [15]. The intraclass correlation coefficients ICC_(2,k)_ and ICC_(3,k)_, were used to evaluate inter-rater and test–retest reliabilities of the Thai-HFHAT-SF, respectively. Reliability results were classified into 4 levels: an ICC < 0.50 was classified as having poor reliability, 0.50–0.74 as moderate, 0.75–0.90 as good, and >0.90 as excellent reliability [16].

## 3. Results

### 3.1. Phase I

The CFA of the 69-question Thai-HFHAT was performed. The results of the analysis were categorized into two subsections: the results of assumption tests and the results of CFA.

#### 3.1.1. Results of Assumption Tests

Univariate outlier and multivariate outlier tests were performed to prevent misinterpretation of the results. Univariate outlier values were used to calculate standardized scores (|z| ≥ 3.30), whereas multivariate outlier values were used for finding Mahalanobis distance (*p* < 0.001). Results showed that no study subjects exhibited a z score of ≥3.30 as an outlier, but 8 of the total subjects had Mahalanobis distance (*p* < 0.001). We determined to continue our study with a total number of 442 study subjects. Data distribution of each question listed in 9 sections of the Thai-HFHAT was examined and a normal distribution was indicated by the graph depicting a bell curve. Maximum likelihood estimation was conducted in the confirmatory factor analysis based on skewness of 2 or less and kurtosis of 7 or less [17]. Our results showed that skewness values were in a range of −0.110–1.385 and kurtosis was in a range of −1.886–0.896. The Kaiser–Meyer–Olkin measure of sampling adequacy was also conducted to determine the suitability or adequacy of data to be used for factor analysis [18]. Results from the measurement indicated that our data was suited for the analysis (0.679). Bartlett’s test of sphericity was conducted in the phase I study to test the hypothesis that the correlation matrix was an identity matrix, which showed a true association between questions. Chi-square was used to interpret the results (*p* < 0.05). Results showed that *χ*^2^ was 466.54 (*p* < 0.001), indicating that the set of data could be used for confirmatory factor analysis.

#### 3.1.2. Results of the CFA

Results of the secondary order CFA of the home fall hazards risk model showed that component weights of the four factor groups had positive values, ranging from 0.31 to 0.67, but no statistical significance (<0.05) was observed. Weights were as follows: 0.67, 0.60, 0.32, and 0.31 with factor loadings ranging from 0.10 to 0.71. The fitness index showed values of *χ*^2^/*df* = 1.38, GFI = 0.988, AGFI = 0.970, SRMR = 0.030, and RMSEA = 0.029, indicating that the model was fit (Figure 1 and Table 1).

### 3.2. Phase II

The demographic characteristics of 150 subjects are shown in Table 2. Forty-four percent and 42% of the elderly had hypertension and hyperlipidemia, respectively. Fifty-two percent of the caregivers had an intimate relationship with the older person. The mean (±SD) duration of caregiving was 20.65 (±3.54) h/day or 5.28 (±0.52) days/week. The mean (±SD) duration of working experience of the VHV subjects was 10.74 (±4.13) years.

#### 3.2.1. Thai Home Falls Hazards Assessment Tool-Short Form (Thai-HFHAT-SF) Content Validity

Twenty-two out of the 29 items on the Thai-HFHAT had excellent agreement (I-CVI = 1) and 7 items had acceptable agreement (I-CVI = 0.7). The content validity for the scale of the instrument was good (S-CVI = 0.9). The value of I-CVI and S-CVI are shown in Table 3.

#### 3.2.2. Inter-Rater and Test–Retest Reliability

The ICC of inter-rater reliability for the Thai-HFHAT-SF was 0.82 (95% CI: 0.71–0.89). The mean (±SD) scores by the elderly, caregiver, and VHV groups were 4.56 (±3.29), 4.23 (±3.22), and 3.35 (±2.65), respectively, as shown in Figure 2. The ICC of test–retest reliability was 0.77 (95% CI: 0.60–0.87) for the elderly, 0.85 (95% CI: 0.73–0.91) for the caregivers, and 0.60 (95% CI: 0.29–0.77) for the VHVs. The average scores for the 1st and 2nd visits of the elderly, caregivers, and VHV is shown in Figure 2.

## 4. Discussion

The main purpose of this study was to develop the Thai-HFHAT-SF. The results showed that the tool was reduced to 28 questions in four factor groups. Therefore, the Thai-HFHAT-SF was shorter, took less time, and was more precise when compared to the original version (69-question Thai-HFHAT) [19] and consistent in terms of language and specific contexts to where the instrument would be used for fall risk screening in Thailand when compared with other foreign fall risk assessment tools [6,7].

The item content validity of the Thai-HFHAT-SF was evaluated and 90% of the instrument’s question items had an excellent agreement (I-CVI = 1.00). However, 10% had an acceptable agreement (I-CVI = 0.67) because the opinion of an architectural expert showed that the definition of home hazards represented the risk to overall health, not merely hazards that cause the falls. For example, the item “There is no bright light for activities in the garage” was rated a score of 2 with the reasoning that more items should be added to cover the size of the garage’s area that could cause injuries to the elderly while getting in/out of the car. Nevertheless, the content validity for the scale of the Thai-HFHT-SF was good (S-CVI, 0.90).

The inter-rater reliability for the Thai-HFHAT-SF was 0.82. The findings were consistent with the previous study that examined the inter-rater reliability of the 69-question Thai-HFHAT, showing ICC = 0.87 [9]. Inter-rater reliability for the Thai-HFHAT-SF had an ICC lower than that of the Thai-HFHAT because the latter only focused on 30 elderly and 1 VHV with moderate reliability of the raters [9]. In addition, the inter-rater reliability of Thai-HFHAT-SF was higher than the 87-question HOME FAST-SR [20] because the Thai-HFHAT-SF was the condensed structure [21]. Therefore, the Thai-HFHAT-SF could be easily filled out, had fewer items, and used less time to complete than the Thai-HFHAT version.

The mean score of the Thai-HFHAT-SF was highest in the elderly group. This was due to most of the elderly having health issues with falling being one of the health issues posing a serious threat to their health [22,23], causing them to sensitively focus on home hazards that could potentially cause fall risk while taking the instrument. The mean score in the VHV group was the lowest, which was consistent with findings reported by Morgan and coworkers (2005). The VHV group may not pay special attention to some of the items that were hazardous in the eyes of the other two groups. Of all the questions, the item “There is a bright light suitable for daily activities” may be overlooked, and the hazard entailing that question may be disregarded by the VHV group. On the other hand, the older person subjects considered that question seriously, and they tended to view the bright light in their home as a potential hazard that could increase fall risk [24,25].

The ICC of the test–retest reliability was 0.77, which was consistent with the previous study conducted with the 69-question Thai-HFHAT with an ICC of test–retest reliability of 0.78 [9]. The result was also consistent with the findings from Vu and Mackenzie [16] and Romli et al. [26] with an ICC of 0.77 and 0.88, respectively. However, the ICC value of the test–retest reliability of the HOME FAST-SR was inconsistent (ICC = 0.71) [20], which may contribute to the larger amount of questions on the instrument that took much longer for the elderly to complete. The Thai-HFHAT-SF was therefore acceptable as a fall risk screening instrument for Thais.

The means scores for the Thai-HFHAT-SF by the elderly group, the caregiver group, and the VHV group were slightly different between the 1st and 2nd home visits. This may contribute to the environment in the homes of the elderly or behavior changes of the elderly after the first home visit. As the elderly completed the first-time rating of the tool, they may pay more attention to their surroundings and make some changes to the environment to make their home safer [27]. This phenomenon is called reactivity and can happen to study subjects when the instrument is administered multiple times [28]. The mean of ratings by the caregiver group and the VHV group was higher from the first-time visit. This was possible because both groups of the raters considered that the change made to the home environment did not affect home hazards that the elderly had to encounter.

The question items of the Thai-HFHAT-SF were reduced to 28. The repetitive questions about lighting, slippery floors, or uneven ground were found when assessing each room in the Thai-HFHAT version. The findings in this study also confirmed that the 28-question Thai-HFHAT-SF had good inter-rater reliability, making this instrument acceptable as a reliable fall risk instrument for Thais. We recommend the Thai-HFHAT-SF for environmental assessment; however, fall prevention should cover both environmental and biological assessment. Kim et al. reported that the new fall risk assessment (FRA) system may be used to evaluate the biological hazards because of its moderate to excellent inter-rater reliability and sensitivity to change in community-dwelling older adults [29]. The limitations of this study were: (1) The data cannot be applied to other study populations because they were collected from only one community, (2) asking the elderly group to inform us of the fall history within the past six months before participating in this study may cause recall bias may occur, and (3) using convenient sampling to include participants may not be able to include all types of the home.

## 5. Conclusions

The inter-rater reliabilities of the HOME FAST-SR and Thai-HFHAT were moderate and good, respectively. The test–retest reliability of the 28-question Thai-HFHAT-SF was good. The instrument appears to be a reliable instrument for identifying home hazards that pose an increased risk for falls, and for screening the elderly who have potential risk for falls in the study population. We suggest that prospective studies are needed among various populations to ensure that this instrument can be applied to the majority of the elderly in Thailand and used for fall prediction in all types of Thai homes.

## Figures and Tables

**Figure 1 ijerph-19-05187-f001:**
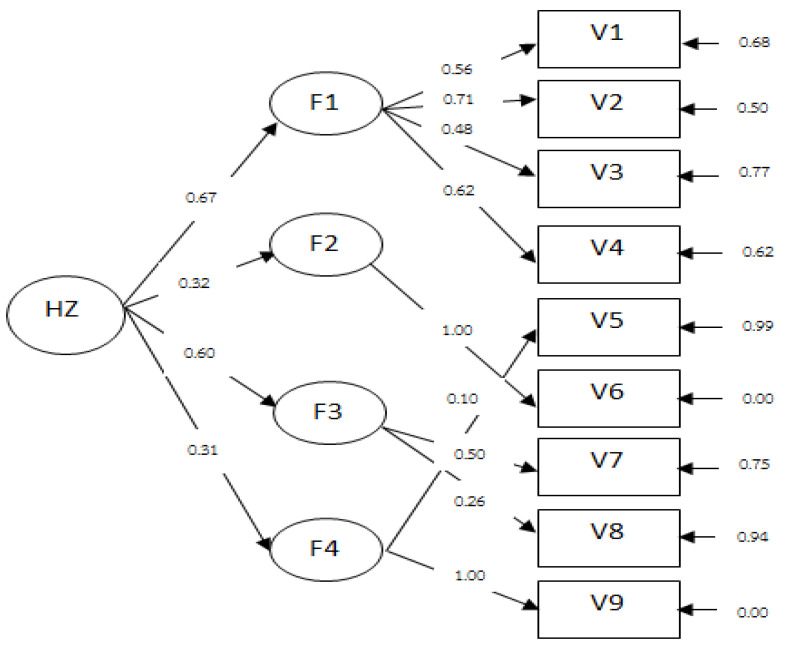
The model of home fall hazards risk. Four factors consisted of F1—indoor area, F2—garage, F3—outdoor area, and F4—risky spots/areas including pets.

**Figure 2 ijerph-19-05187-f002:**
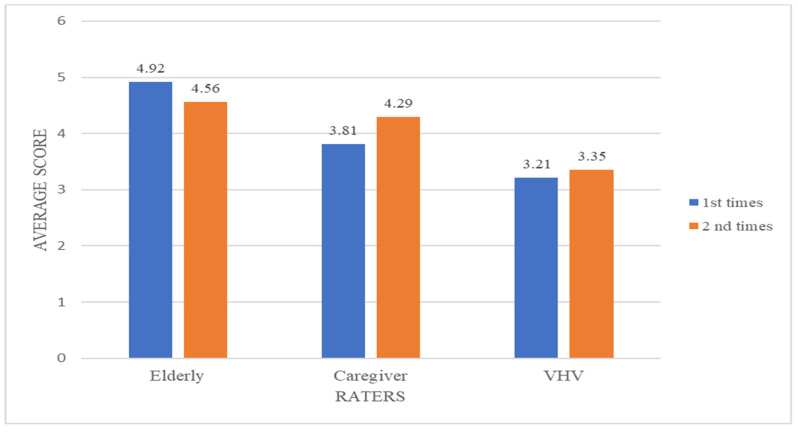
The average scores from the elderly caregivers and VHVs for the 1st and 2nd visit.

**Table 1 ijerph-19-05187-t001:** The factors and the items of the Thai-HFHAT-SF.

No	Factors	Items
**F1**	**Indoor area**	The lighting is not suitable for activities Slippery/unsmooth surface The steps in the room Cluttered objects or wires block the path Unused mat/rug/cloth not firmly attached to the floor Cabinets are too low or too high Cannot turn on the light from the bed Lying on the floor The bathroom is located outside the home No handrails in the bathroom No shower seat/shower chair Do not use the toilet/toilet bowl with hanging legs
**F2**	**Garage**	The lighting is not suitable for activities Slippery/unsmooth surface The steps in the garage Cabinets are too low or too high
**F3**	**Outdoor area**	The lighting is not suitable for activities The steps around the outdoor area The corridors around the home are not in good condition Slippery stair surface The stairs are too steep Lack of handrails/handrails are not strong Wearing inappropriate shoes
**F4**	**Risky spots/areas including pets**	**Stairs in the home**
		Slippery stair surface Cluttered objects block the path The stairs are too steep Lack of handrails/handrails are not strong
		**Pets**
		5. Having pets in the house that pose a risk of falls

**Table 2 ijerph-19-05187-t002:** Demographic characteristics of participants; elderly (*n* = 50), caregiver (*n* = 50), and VHV (*n* = 5).

Characteristics	Elderly	Caregiver	VHV
	*n* (%)	*n* (%)	*n* (%)
**Sex**			
Male	19 (38.00)	22 (44.00)	-
Female	31 (62.00)	28 (56.00)	5 (100)
Mean (SD) age in years	73.46 (6.70)	58.62 (15.84)	45.5 (6.18)
Mean (SD) BMI in kg/m^2^	22.29 (4.39)	22.51 (3.70)	22.33 (4.43)
**Education level**			
Grades 1–3	42 (84.00)	18 (36.00)	1 (20.00)
>Grades 3	8 (16.00)	32 (64.00)	4 (80.00)
**Marital status**			
No partner	6 (12.00)	6 (12.00)	1 (20.00)
Have a partner	44 (88.00)	44 (88.00)	4 (80.00)
**Occupations**			
None	13 (26.00)	7 (14.00)	0 (0.00)
Agriculture	12 (24.00)	22 (44.00)	2 (40.00)
Housekeeper	16 (32.00)	6 (12.00)	2 (40.00)
Personal business	7 (14.00)	7 (14.00)	1 (20.00)
Official	2 (4.00)	3 (6.00)	0 (0.00)
Others	0 (0.00)	5 (10.00)	0 (0.00)
**History of falling in the previous 6 months of the older person subjects**			
Yes	14 (28.00)	-	-
No	36 (72.00)	-	-
**The cause of falling**			
Tripping	8 (57.14)	-	-
Loss of balance	5 (35.72)	-	-
Slip	1 (7.14)	-	-

SD, standard deviation. *n*, number.

**Table 3 ijerph-19-05187-t003:** The value of the item content validity index (I-CVI) and the content validity for the scale (S-CVI).

No	Groups	Items	I-CVI
1	**Indoor area**	The lighting is not suitable for activities	1
2		Slippery/unsmooth surface	1
3		The steps in the room	0.7
4		Cluttered objects or wires block the path	1
5		Unused mat/rug/cloth not firmly attached to the floor	1
6		Cabinets are too low or too high	1
7		Cannot turn on the light from the bed	1
8		Lying on the floor	1
9		The bathroom is located outside the home	1
10		No handrails in the bathroom	1
11		No shower seat/shower chair	0.7
12		Do not use the toilet/toilet bowl with hanging legs	1
13	**Garage**	The lighting is not suitable for activities	1
14		Slippery/unsmooth surface	1
15		The steps in the garage	0.7
16		Cabinets are too low or too high	1
17	**Outdoor area**	The lighting is not suitable for activities	0.7
18		The steps around the outdoor area	1
19		The corridors around the home are not in good condition	1
20		Slippery stair surface	1
21		The stairs are too steep	0.7
22		Lack of handrails/handrails are not strong	1
23		Wearing inappropriate shoes	0.7
24	**Risky spots/** **areas including pets**	Slippery stair surface	1
25		Cluttered objects block the path	1
26		The stairs are too steep	0.7
27		Lack of handrails/handrails are not strong	1
28		Having pets in the house that pose a risk of falls	1
**S-CVI**			**0.9**

## Data Availability

Not applicable.

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
