# Peer review of "Construction of the Short-Form Thai-Home Fall Hazard Assessment Tool (Thai-HFHAT-SF) and Testing Its Validity and Reliability in the Elderly"

_ijerph, 2022, doi:10.3390/ijerph19095187_

Round 1

Reviewer 1 Report

I am honored to review this work. The authors have done commendable works in the review. Overall, it is technically sound but still requires a little editorial formatting.
Find below my specific comments.
- Abstract: Some therms need to be more clear to justify the importance of this work
- Keywords: They need to be standardized according to the Mesh Descriptor, or according to the paper guidelines
- Table 2: A misspelling (Aagriculture). 
- Figure 2: The first group is named "Elderly", and the whole paper used "older persons". I extremly recommend the therm "Elderly" for both figure and paper.
- I reccomend an English review for minor grammar errors.

Author Response

We changed the abbreviations of the terminology to be more clear that shown on lines 27 to 30 in the Abstract section.

We used Mesh Descriptor to change the keywords from

- “home hazards” to “home environments”

- “Older persons” to “elderly”

, and cut off “Thai-HFHAT”

We corrected the misspelled words in Table 2 from “Aagriculture” to “Agriculture”

We changed the words from “older persons” to “elderly” in both figure and paper including in the article title.

We already submitted a language validation with International Journal of Environmental Research and Public Health already.

Reviewer 2 Report

The paper need a minor revision of introduction. The topic of falls is of great relevance and authors could add other evidence on its impact on the health of the elderly and on the importance of preventing them. From this point of view the instrument takes on even more .

Author Response

We added more evidence of the impact of falling on the health of the elderly and on the importance of preventing falls on line 44 to 50.

Reviewer 3 Report

A short format of fall risk assessment instrument was developed for use in the elderly in their home environment and showed validity and reliability, and rapid application, being adequate for the assessment of domestic risk of elderly people in Thailand. A relevant and current article, with a good design and that brings innovation to science, but requires detailing how the inclusion of participants in the study was performed through recruitment/ volunteering and include in the study the limitations of this type of sampling.

Author Response

  • The inclusion of participants in the reliability phase of this study using a convenient method is shown on line 118 to 120.
  • The limitation of the convenient sampling method is shown on line 345 to 346 in the Discussion section.

Reviewer 4 Report

1. instruction
There is a lack of justification for the necessity of your research. You need to convince yourself why you need this tool for falls. Of course, I understood the need for a short tool.
2. method
  A rationale for the number of personnel recruited in step 1 is required. In addition, a clear presentation and rationale for the number of CFA and EFA personnel will be required.
In addition, a detailed explanation of the overall tool development process is required.
3. Results
It is necessary to present a table for the content validity index.
Provide the cronbach'a value, which is the confidence value.
4. Discussion
Comparison with the original measurement tool is necessary, and comparison with other fall risk assessment tools is required. I would like to explain the significance of my research in various aspects.
5. conculsion
Please explain what your research suggests.

Author Response

We added a reason for needing Thai-HFHAT for screening falls in the elderly on line 52 to 54 and on line 69 to 70.

-              We added a rationale for the number of personnel recruited in step1, in CFA on line 90 to 92 in the Materials and Method section.

-              We added a detailed explanation of the overall tool development process including 69-question Thai-HFHAT on line 92 to 99 and 28-question Thai-HFHAT on line 102-110 in the Materials and Method section.

We added the table for the content validity index that is shown in Table 3 in the Result section.

We think that Cronbach 'a value is used to present the confidence of the questionnaire. But the tools shown in this paper are home hazard assessment tools, that explore the risk points that are the cause of falls in the home, which is a different fact in each participant's home.  As well as the development of the original version (69-question Thai-HFHAT) and other widely used risk assessment tools, Cronbach 'a value was not shown for the confidence of the tools.

Reference:

1)            Lektip, C.; Rattananupong, T.; Sirisuk, K.; Suttanon, P.; Jiamjarasrangsi, W. Adaptation and evaluation of home fall risk assessment tools for the elderly in Thailand. Southeast Asian J. Trop. Med. Public Health. 2020, 51, 65–76.

2)            Romli MH, Mackenzie L, Lovarini M, Tan MP, Clemson L. The clinimetric properties of instruments measuring home hazards for older people at risk of falling: a systematic review. Eval Health Prof 2018; 41: 82-128.

We added the comparison with the original measurement tool and with other fall risk assessment tools on line 277 to 282.

We already added the research suggestion on line 353 to 355 in the Conclusion section.

Round 2

Reviewer 4 Report

Thank you so much
Best wishes for your future.